# Vitamin D_3_ Upregulated Protein 1 Deficiency Promotes Azoxymethane/Dextran Sulfate Sodium-Induced Colorectal Carcinogenesis in Mice

**DOI:** 10.3390/cancers16172934

**Published:** 2024-08-23

**Authors:** Ki Hwan Park, Hyoung-Chin Kim, Young-Suk Won, Won Kee Yoon, Inpyo Choi, Sang-Bae Han, Jong Soon Kang

**Affiliations:** 1Laboratory Animal Resource Center, Korea Research Institute of Bioscience and Biotechnology, 30 Yeongudanji-ro, Cheongwon-gu, Cheongju-si 28116, Chungcheongbuk-do, Republic of Korea; brightnessd@kribb.re.kr (K.H.P.); hckim@kribb.re.kr (H.-C.K.); yswon@kribb.re.kr (Y.-S.W.); wkyoon@kribb.re.kr (W.K.Y.); 2Immunotherapy Research Center, Korea Research Institute of Bioscience and Biotechnology, 125 Gwahak-ro, Yuseoung-gu, Daejeon-si 34141, Republic of Korea; ipchoi@kribb.re.kr; 3College of Pharmacy, Chungbuk National University, 194-21 Osongsaemgmyung-1-ro, Heungdeok-gu, Cheongju-si 28160, Chungcheongbuk-do, Republic of Korea; shan@chungbuk.ac.kr

**Keywords:** inflammatory bowel disease, colitis-associated colorectal cancer, vitamin D_3_ upregulated protein 1

## Abstract

**Simple Summary:**

Inflammatory bowel disease (IBD) increases the risk of developing colitis-associated colorectal cancer (CAC) compared to the general population. The specific genetic alterations associated with the onset of CAC are largely unknown. In this study, we investigated the role of vitamin D3 upregulated protein 1 (VDUP1) in a CAC mouse model. VDUP1 knockout (KO) increased the severity of colitis and CAC development. Our data suggest that VDUP1 may be explored as a potential therapeutic approach for CAC prevention.

**Abstract:**

VDUP1 acts as a tumor suppressor gene in various cancers. VDUP1 is expressed at low levels in sporadic and ulcerative-colitis-associated colorectal cancer. However, the effects of *VDUP1* deficiency on CAC remain unclear. In this study, we found that *VDUP1* deficiency promoted CAC development in mice. Wild-type (WT) and *VDUP1* KO mice were used to investigate the role of VDUP1 in the development of azoxymethane (AOM)- and dextran sulfate sodium (DSS)-induced CAC. VDUP1 levels significantly decreased in the colonic tumor and adjacent nontumoral tissues of WT mice after AOM/DSS treatment. Moreover, AOM/DSS-treated *VDUP1* KO mice exhibited a worse survival rate, disease activity index, and tumor burden than WT mice. *VDUP1* deficiency significantly induced cell proliferation and anti-apoptosis in tumor tissues of *VDUP1* KO mice compared to WT littermates. Additionally, mRNA levels of interleukin-6 and tumor necrosis factor-alpha and active forms of signal transducer and activator of transcription 3 and nuclear factor-kappa B p65 were significantly increased in the tumor tissues of *VDUP1* KO mice. Overall, this study demonstrated that the loss of *VDUP1* promoted AOM/DSS-induced colon tumorigenesis in mice, highlighting the potential of VDUP1-targeting strategies for colon cancer prevention and treatment.

## 1. Introduction

GLOBOCAN 2020 data estimated colorectal cancer (CRC) as the third most common cancer in men and women and second leading cause of cancer-related death worldwide [1]. The age-standardized incidence rate of CRC is 23.4 per 100,000 people in men and 16.2 in women, with corresponding age-mortality rates of 11.0 and 7.2 per 100,000 population, respectively [2]. Chronic inflammation, as observed in inflammatory bowel disease, significantly increases the risk of CRC development [3,4,5,6]. However, specific genetic alterations associated with colitis-associated CRC (CAC) onset and their effects remain unknown. 

Uncontrolled cell proliferation and suppression of apoptosis are essential processes in the pathogeneses of various malignancies, including CRC. Aberrant activation of Wnt signaling induces cell proliferation and inhibits cell death. Prolonged activation of the Wnt signaling pathway promotes the accumulation and translocation of β-catenin from the cell membrane to the cytoplasm and nucleus, resulting in excessive activation of the oncogenic target genes promoting tumor cell proliferation and survival [7]. Gain-of-function mutations in TP53 have been observed in CAC, which contribute to increased tumor invasiveness, suppression of apoptosis, and increased genomic instability [8,9]. Treatment with azoxymethane (AOM)/dextran sulfate sodium (DSS) in p53 knockout (KO) mice results in increased tumor formation and nuclear localization of β-catenin, accompanied by excessive activation of Wnt-responsive genes [10]. Abnormal activation of β-catenin is indicated by the significant increase in cyclin D1 and decrease in cleaved caspase-3 levels in CAC [11].

Tumor necrosis factor-alpha (TNF-α) and interleukin (IL)-6 levels are elevated in the tumor tissues and sera and associated with poor survival, increased tumor burden, and disease progression in patients with CRC [12,13]. Moreover, TNF-α and IL-6 promote the activation of key oncogenic transcription factors, including nuclear factor-kappa B (NF-κB) and signal transducer and activator of transcription 3 (STAT3), that contribute to the development and progression of CRC, particularly CAC [14,15,16]. *TNF*-*α* [17] or *IL*-*6* [15] KO significantly inhibits CAC development in an AOM/DSS model, highlighting the importance of these findings from a translational research perspective. Therefore, many studies have investigated the mechanisms by which CAC onset is influenced by intricate interactions among signaling molecules, particularly those within the STAT3 and NF-κB pathways, by examining the impact of overexpression or suppression of specific genes in inflammatory and carcinogenic environments on these signaling cascades [18,19,20,21,22,23].

Vitamin D_3_ upregulated protein 1 (VDUP1), also known as the thioredoxin-interacting protein or thioredoxin-binding protein-2, was initially identified in HL-60 cells treated with 1,25-dihydroxy vitamin D_3_ [24]. *VDUP1* acts as a tumor suppressor gene (TSG) and is significantly downregulated in various cancers via epigenetic and genetic mechanisms [25,26]. Loss of *VDUP1* is involved in carcinogenesis, including gastric [27], bladder [28], and hepatocellular [29] carcinogenesis. Several studies have suggested the impact of VDUP1 on CRC prognosis. Kato et al. demonstrated that high RET finger protein expression correlates with the downregulation of VDUP1 expression and poor clinical outcomes in human colon cancer [30]. Hu et al. reported that VDUP1 plays a critical role in the tumor suppressor function of N-myc downstream-regulated gene 2 in Caco-2 and HT-29 cells [31]. Furthermore, VDUP1 expression is decreased in human ulcerative colitis (UC) and CRC [32].

Therefore, we explored the function of VDUP1 in the development of DSS-induced acute UC in mice [33]. Our previous studies demonstrated that *VDUP1* deficiency in a DSS-induced colitis model leads to severe tissue damage in the colon, accompanied by marked NF-κB activation, inflammation, and increased macrophage chemotaxis. However, the role of VDUP1 on CAC development remains unclear. In this study, we investigated the effect of *VDUP1* deficiency on CAC development using *VDUP1* KO and AOM/DSS mouse model. We demonstrated that decreased VDUP1 expression in human and mouse CRC correlates with poor prognosis. We also provided genetic and biochemical evidence that the loss of *VDUP1* is critical for tumorigenesis in CAC model mice. This study revealed the critical role of VDUP1 in CAC development, suggesting it as a potential therapeutic target for CAC.

## 2. Materials and Methods

### 2.1. Reagents and Animals

Unless otherwise specified, reagents were obtained from Sigma-Aldrich (St. Louis, MO, USA). *VDUP1*-KO mice were produced according to the method described previously [34]. Briefly, mice were kept on a C57BL/6 background and were backcrossed with C57BL/6 mice for over 10 generations. The mice used in this study were maintained under specific pathogen-free conditions. The animal study was conducted with approval from the Institutional Animal Care and Use Committee at the Korea Research Institute of Bioscience and Biotechnology (approval no. KRIBB-AEC-13165).

### 2.2. AOM/DSS-Induced Colitis

To establish AOM/DSS-induced CAC model mice, 7-week-old male mice were administered with 7.4 mg/kg AOM on day 0. After seven days, 2% DSS (*w*/*v*) (35–55 kDa, TdB Consultancy; Uppsala, Sweden) was administered with drinking water for four days, followed by regular water for 17 days. Three cycles of DSS treatment were performed. Body weight, stool consistency, and stool bleeding were recorded weekly. The disease activity index parameters, including weight loss, stool consistency, and rectal bleeding, were assessed based on the specified criteria (Appendix A). Animal survival was monitored daily. On day 85, the mice were euthanized and the colons and tumors were harvested for further analyses.

### 2.3. RNA Isolation and Quantification of mRNA Expression

Total RNA was extracted from the colon and tumor tissues using TRIzol reagent (Ambion, Austin, TX, USA) according to the manufacturer’s protocol. Briefly, tissues were collected and homogenized using the FastPrep^®^-24 (MP Biomedicals, Solon, OH, USA) and TRIzol reagent at a speed of 5.0 m per second (m/s) for 30 s. Chloroform was added to the homogenate and mixed thoroughly. The mixture was centrifuged at 12,000× *g* for 15 min at 4 °C to separate the phases, and the aqueous phase containing RNA was transferred to a new tube. RNA was precipitated by adding isopropanol. The mixture was centrifuged at 12,000× *g* for 10 min at 4 °C. The RNA pellet was then washed with ethanol, air dried, and dissolved in RNase-free water. RNA quality and quantity were assessed by measuring absorbance at 260 and 280 nm using infiniteM200 (TECAN, Männedorf, Switzerland). cDNA was synthesized from 1 μg of total RNA through reverse transcription using the AccuPower RT PreMix (Bioneer, Daejeon, Republic of Korea) according to the manufacturer’s instructions at 42 °C for 60 min and 95 °C for 5 min. The cDNA was then subjected to a quantitative reverse transcription–polymerase chain reaction (qRT-PCR) using the Power SYBR Green PCR Master Mix (Invitrogen, Carlsbad, CA, USA) in a thermal cycler. qRT-PCR was performed with 45 cycles of denaturation at 95 °C for 15 s and amplification at 60 °C for 1 min using the ABI 7500 Fast Real-Time PCR System (Applied Biosciences, Foster City, CA, USA). Relative gene expression levels were calculated using the 2^−△△Ct^ method, normalized to β-actin. Primer sequences are listed in Appendix A.

### 2.4. Histology, Immunohistochemistry (IHC), and TdT-Mediated dUTP Nick-End Labeling (TUNEL) Assay

Freshly collected colons and tumors were washed with 1X phosphate-buffered saline and fixed in 10% buffered formalin. The fixed tissues were than embedded in paraffin. Subsequently, paraffin-embedded tissues were sectioned, mounted on slides, and stained with hematoxylin and eosin (H&E). H&E-stained tissue sections were graded in a blinded manner. IHC was conducted using an avidin–biotin complex staining kit (Vector Laboratories, Burlingame, CA, USA). Following incubation with specific antibodies, the sections were developed with the DAB Substrate Kit (Vector Laboratories, Burlingame, CA, USA). All antibodies used in this study are listed in Appendix A. Finally, the apoptotic cells were detected via the TUNEL Apoptosis Detection Kit (Millipore, Billerica, MA, USA) according to the manufacturer’s instructions.

### 2.5. Western Immunoblotting Analysis

Total protein was extracted by lysing the tissues in the Cell Lysis Buffer (Cell Signaling Technology, Beverly, MA, USA) with a phosphatase inhibitor (Sigma-Aldrich) and a protease inhibitor cocktail (Merck Millipore, Billerica, MA, USA). The protein concentration in the lysate was measured using the Bradford protein assay kit (Bio-Rad, Hercules, CA, USA) according to the manufacturer’s protocols. Proteins were separated using sodium dodecyl sulfate (SDS)-polyacrylamide gel electrophoresis with running buffer (25 mM Tris, 192 mM glycine, and 0.1% SDS) and transferred to polyvinylidene difluoride membranes using transfer buffer (25 mM Tris Base, 192 mM glycin, and 20% methanol). The membranes were then incubated with a blocking buffer consisting of Tris-buffered saline (TBS) containing 0.05% Tween 20 and 5% non-fat dried milk for 1 h at room temperature. Subsequently, the membranes were washed three times with TBS containing 0.05% Tween 20 (TBST) for 10 min each wash at room temperature. The membranes were probed with the indicated primary antibodies (Appendix A) diluted in TBST. Following incubation with primary antibodies, the membranes were washed three times with TBST for 10 min each wash. Subsequently, the membranes were probed with horseradish peroxidase (HRP)-conjugated secondary antibody for 1 h at room temperature. After secondary antibody incubation, the membranes were washed as previously described. Protein bands were visualized using the Immobilon Western Chemiluminescent HRP Substrate (Merck Millipore). Band intensities were quantified using the ImageJ software bundled with 64-bit Java 8 (National Institutes of Health, Bethesda, MD, USA) according to the user guide and normalized to β-actin levels.

### 2.6. Statistical Analyses

Data are presented as the mean ± standard error of the mean. Statistical evaluation was performed using one- and two-way ANOVA with Tukey’s multiple comparisons test using the GraphPad Prism software 10.0.2 (GraphPad Software, La Jolla, CA, USA). The Log-rank (Mantel-Cox) test was employed to compare the survival. Statistical significance was defined as * *p* < 0.05, ** *p* < 0.01, *** *p* < 0.001, **** *p* < 0.0001, and not significant (ns).

## 3. Results

### 3.1. VDUP1 Expression Is Downregulated and Correlates with Poor Prognosis in CAC

In our previous study of acute DSS-induced colitis, we found that *VDUP1* KO mice displayed more intense colitis compared to WT mice, attributed to increased macrophage infiltration and inflammation in the intestinal tissues [33]. In this study, we evaluated the extent of colorectal tumorigenesis in *VDUP1* KO mice using an AOM/DSS model. To assess the clinical significance of VDUP1 in CRC and UC-associated CRC, we conducted bioinformatics analysis using public data. TCGA-COAD data, consisting of 275 CRC and 45 normal colon tissues, revealed significantly lower VDUP1 expression levels in CRC tissues than in normal tissues (*p* < 0.05; Figure 1A). Additionally, analysis of the GSE3692 dataset revealed significantly lower VDUP1 expression levels in UC-associated cancer tissues than in UC-associated dysplasia (*p* < 0.01) and nontumor (*p* < 0.000) tissues (Figure 1B). Furthermore, low VDUP1 expression was associated with poor OS and disease-specific survival compared to high VDUP1 expression, as indicated by the Sidra-LUMC AC-ICAM dataset via cBioPortal (*p* = 0.0017) and GSE38832 cohort dataset via shinyGEO (*p* = 0.0792), respectively (Figure 1C,D). Consequently, we investigated whether VDUP1 expression was reduced during CAC development using an AOM/DSS-induced mouse model of CAC, as previously described [35]. *VDUP1* mRNA levels were significantly decreased in both the tumor (*p* < 0.0001) and nontumor (*p* < 0.0001) tissues following AOM/DSS treatment (Figure 1E). Moreover, VDUP1 protein levels were decreased in both the tumor and nontumor tissues of AOM/DSS-treated mice compared to those in control mice (Figure 1F). These results suggest that VDUP1 is involved in CAC onset.

### 3.2. VDUP1 Reduces Disease Severity in the AOM/DSS-Induced CAC Model Mice

To determine whether VDUP1 protects against chronic inflammation-associated carcinogenesis, we challenged WT and *VDUP1* KO mice with AOM/DSS. *VDUP1* KO mice exhibited greater sensitivity to AOM/DSS than WT mice, as indicated by decreased survival (*p* < 0.0001; Figure 2B, Appendix A). Indeed, only 50% of *VDUP1* KO mice survived the entire treatment regimen compared to 100% of WT mice. We also monitored the colitis symptoms, including changes in body weight (Figure 2C), rectal bleeding, and stool consistency, according to the experimental schedule (Figure 2A). AOM/DSS-treated *VDUP1* KO mice exhibited a significant early enhancement in the disease activity index (*p* < 0.0001; Figure 2D), which is a cumulative score based on body weight loss (Appendix A), rectal bleeding (Appendix A), and stool consistency (Appendix A), throughout each DSS cycle compared with similarly treated WT mice (*p* < 0.0001; Figure 2D).

### 3.3. VDUP1 Protects against AOM/DSS-Mediated Carcinogenesis

*VDUP1* KO mice exhibited significantly decreased colon length after AOM/DSS treatment compared to WT mice, with a mean change in colon length of −1.126 cm in *VDUP1* KO mice versus −0.2881 cm in WT mice (*p* < 0.001; Figure 3A,B). The colon length of *VDUP1* KO mice was similar to that of WT mice (Appendix A). *VDUP1* KO mice showed significantly increased tumor number (Figure 3C) and size (Figure 3D) after AOM/DSS treatment compared to WT mice, with a mean tumor number of 7.1 in *VDUP1* KO mice versus 2.8 in WT mice (*p* < 0.0001; Figure 3C) and mean of tumor size of 10.6 mm^2^ in *VDUP1* KO mice versus 6.8 mm^2^ in WT mice (*p* < 0.001; Figure 3D). Moreover, *VDUP1* KO mice showed significantly enhanced tumor grade after AOM/DSS treatment compared to WT mice (Figure 3E,F). *VDUP1* KO mice also exhibited a slightly higher number of adenomas with low-grade dysplasia (ADL) and high-grade dysplasia (ADH) than WT mice. The mean ADL and ADH were 3.6 and 4.8 in *VDUP1* KO mice versus 2.2 and 1.4 in WT mice, respectively (Figure 3F). However, *VDUP1* KO mice exhibited a significantly higher number of adenocarcinomas (AD) than WT mice, with a mean AD of 4.8 in *VDUP1* KO mice versus 0.2 in WT mice (*p* < 0.0001; Figure 3F).

### 3.4. VDUP1 Deficiency Induces Cell Proliferation and Inhibits Apoptosis in AOM/DSS-Induced CAC Model Mice

To better define the cellular phenotype of tumorigenesis, we stained for Ki-67, a well-recognized marker of hyperproliferation and tumorigenesis [36]. Additionally, to assess the association between VDUP1 and apoptosis in tumor tissues, they were stained for TUNEL staining, a marker of apoptotic cells [37]. The tumor tissues of AOM/DSS-treated *VDUP1* KO mice exhibited a significantly increased number of Ki-67-positive cells compared to those of WT mice (Figure 4A), with a Ki-67 index of 52.1 in *VDUP1* KO mice versus 33.4 in WT mice (*p* < 0.0001; Figure 4C). No significant difference in Ki-67-positive cells was observed in the villi of the control groups between WT and *VDUP1* KO mice (Appendix A). Furthermore, the tumor tissues of AOM/DSS-treated *VDUP1* KO mice showed a significantly decreased number of TUNEL-positive cells compared to those of WT mice (Figure 4B), with an apoptotic index of 6.4 in *VDUP1* KO mice versus 1.7 in WT mice (*p* < 0.0001; Figure 4D).

### 3.5. VDUP1 Deficiency Induces the Molecular Patterns of Carcinogenesis in AOM/DSS-Induced CAC Model Mice

Consistent with the above-mentioned results, IHC data revealed that p53 protein was poorly expressed in the cytosol of *VDUP1* KO mice and highly expressed in the nucleus of WT mice (Figure 5A). Compared to WT mice, *VDUP1* KO mice exhibited a significant increase in β-catenin protein incorporation into the tumor tissues (Figure 5B). Additionally, mRNA analysis revealed significant upregulation of the level of anti-apoptotic gene *Bcl*-XL (Figure 5C) and proliferation-inducing gene cyclin D1 (Figure 5D) in the tumor tissues of *VDUP1* KO mice compared to those of WT mice. Protein expression levels of cleaved caspase-3 (Figure 5E,F), an apoptotic protein, were lower in the tumor tissues of *VDUP1* KO mice than in those of WT mice. However, protein levels of cyclin D1 (Figure 5E,G), a proliferation-inducing protein, were higher in the tumor tissues of *VDUP1* KO mice than in those of WT mice.

### 3.6. VDUP1 Deficiency Induces the Activation of STAT3 and NF-κB in AOM/DSS-Induced CAC Model Mice

IL-6 and TNF-α are significantly upregulated in UC and CRC and closely related to the activation of NF-κB and STAT3, which promote CRC development [38,39,40]. Furthermore, loss of *VDUP1* promotes IL-6, TNF-α, and NF-κB activation in the UC model mice [33]. *VDUP1* deficiency correlates with the activation of NF-κB and STAT3 during tissue regeneration [41]. However, whether the loss of *VDUP1* mediates IL-6 and TNF-α production and activation of NF-κB and STAT3 during carcinogenesis, particularly in CAC, remains unclear. Here, we found that *VDUP1* deficiency significantly increased the mRNA levels of IL-6 and TNF-α in the tumor tissues of AOM/DSS-treated mice compared to those in the tumor tissues of WT mice (Figure 6A,B). Consistently, loss of *VDUP1* significantly increased the nuclear incorporation of the active form of NF-κB p65 (Figure 6C,D) and the protein levels of the active forms of NF-κB p65 and STAT3 (Figure 6E–G).

## 4. Discussion

In this study, we explored the correlation between VDUP1 expression and CAC development. Analysis of mRNA from 7 UC, 10 CRC, and 10 normal colon specimens previously revealed that *VDUP1* mRNA expression was significantly suppressed in UC and CRC tissues compared to that in normal tissues [32]. These observations are consistent with our findings in this study; VDUP1 levels were significantly reduced in the overall CRC specimens but also in the CRC specimens with a history of UC based on data from TCGA-COAD and GSE3629 datasets. Moreover, patients with relatively low VDUP1 expression showed poor clinical outcomes compared to those with high VDUP1 expression based on data from Sidra-LUMC and GSE38832 datasets. Consistent with our analysis of public databases, VDUP1 levels were significantly decreased not only in the tumor tissues but also in the nontumor tissues of AOM/DSS-treated WT mice compared to those in the colon tissues of control mice. Furthermore, *VDUP1* deficiency worsened the clinical symptoms of chronic colitis and decreased the survival rate in a rodent model of CAC. These results suggest the involvement of *VDUP1* deficiency in CAC development, emphasizing its significance from a translational research perspective.

Loss of TSG function in rodent CAC models is characterized by significantly increased cell proliferation and apoptosis inhibition effects in colon tissues, resulting in aggressive tumor formation [42,43,44]. VDUP1 is involved in the induction of cell cycle arrest and apoptosis in cancer cells. We previously demonstrated that the transcriptional activation of VDUP1 by a small activating RNA in the A549 non-small cell lung cancer cell line induces G1 and G2 cell cycle arrest, accompanied by increased protein levels of p53 and p27 [45]. Moreover, upregulation of VDUP1 expression is necessary for apoptotic stimuli, such as dexamethasone and SAHA [46,47]. However, effects of VDUP1 on cell growth inhibition and apoptosis are dependent on the cell type [48]. In this study, *VDUP1* deficiency significantly increased tumor formation in a rodent CAC model. Moreover, loss of *VDUP1* significantly increased the number, size, and grade of tumors. Increased Ki-67 expression and decreased TUNEL-positive cells in tumor tissues reflected the aggressive proliferation and survival of tumor cells in *VDUP1* KO mice after AOM/DSS treatment. Excessive activation of the Wnt/β-catenin pathway induces the neoplastic transformation of epithelial cells and promotes tumor growth, angiogenesis, tumor invasion, and metastasis [49]. Loss of p53 function is associated with the nuclear accumulation of β-catenin and activation of NF-κB and STAT3 in CAC [10,50]. Here, *VDUP1* deficiency increased β-catenin nuclear expression but decreased p53 nuclear expression in tumor tissues after AOM/DSS treatment, along with decreased cleaved caspase-3 and increased Bcl-XL and cyclin D1 levels. These results highlight the antiproliferative and proapoptotic roles of VDUP1 in CAC development.

These results may be attributed to the excessive activation of cytokines and inflammatory transcription factors in chronic colitis. The two most defined proinflammatory and protumor pathways in CAC are the NF-κB and IL-6/STAT3 signaling pathways [40,51,52]. Activation of STAT3 not only induces the expression of cell-proliferation-related genes, including cyclin D1 and proliferating cell nuclear antigen, and antiapoptotic genes, including Bcl-2 and Bcl-XL but also increases the expression levels of proinflammatory and tumor-promoting cytokines, including, IL-6 and TNF-α [53]. Two separate studies have demonstrated that NF-κB [54] and IL-6/STAT3 [15] are essential for CAC development in an AOM/DSS-induced rodent CAC model. Activated STAT3 causes prolonged activation of NF-κB signaling in tumors [55]. Therefore, NF-κB and STAT3 activation should be further explored to obtain more comprehensive insights. Previously, we reported that loss of *VDUP1* significantly increases the mRNA levels of IL-6, TNF-α, IL-1β, and Cox-2 in a rodent model of DSS-induced acute colitis, accompanied by aggressive infiltration of macrophages with NF-κB p65 activation [33]. Consistent with our previous report, we observed significantly increased mRNA levels of TNF-α and IL-6 in the tumor tissues of AOM/DSS-treated *VDUP1* KO mice compared to those in the tissues of WT mice. Moreover, loss of *VDUP1* enhanced the active forms of NF-κB p65 and STAT3, accompanied by a significant increase in the nuclear translocation of phosphorylated p65 to the tumor tissues in CAC model mice compared to that in WT mice. These findings highlight the crucial roles of *VDUP1* deficiency in the activation of NF-κB, STAT3, and downstream protumor mediators in CAC.

## 5. Conclusions

In conclusion, this study demonstrated the significance of the low expression of VDUP1 in CRC using public clinical data and indicated its correlation with poor survival, thus providing important insights for future translational research. Exploration of the mechanisms underlying *VDUP1* deficiency in the experimental CAC model revealed its significant tumor-suppressive roles in CAC. Overall, this study highlights the potential of VDUP1-targeting strategies for CAC treatment.

## Figures and Tables

**Figure 1 cancers-16-02934-f001:**
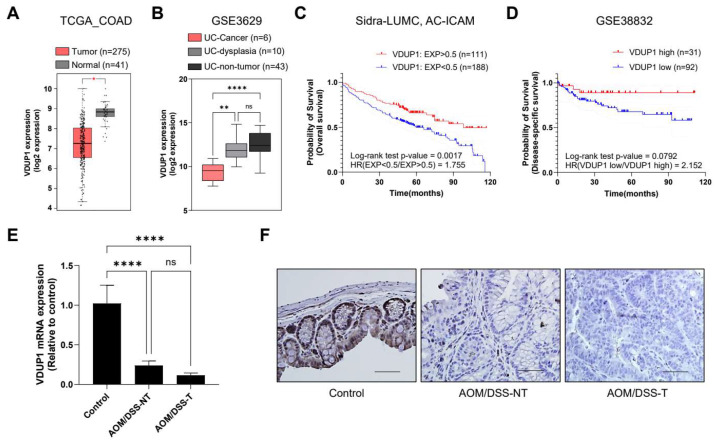
Vitamin D3 upregulated protein 1 (VDUP1) expression is downregulated and associated with poor prognosis in colorectal cancer (CRC). (**A**) Expression levels of VDUP1 in The Cancer Genome Atlas–colon adenocarcinoma (TCGA-COAD) data obtained via the Gene Expression Profiling Interactive Analysis 2 (GEPIA2) server. (**B**) Expression levels of VDUP1 in ulcerative colitis (UC)-cancer, UC-dysplasia, and UC-nontumor GSE3692 data obtained via shinyGEO (*n* = 6, *n* = 10, and *n* = 43, respectively). (**C**) Overall survival (OS) curve of patients with COAD with low (Log2 ratio > 0.05) and high (Log2 ratio < 0.05) VDUP1 transcript levels in Sidra-LUMC AC-ICAM data via cBioPortal (*n* = 229, and *n* = 119, respectively). (**D**) Disease-specific survival curve of patients with COAD with low (cutoff-low, 75%) and high (cutoff-high, 25%) VDUP1 transcript level in GSE38832 data via shinyGEO (*n* = 92, and *n* = 31, respectively). (**E**) *VDUP1* mRNA levels in the control, azoxymethane (AOM)/dextran sulfate sodium (DSS)-nontumor (NT), and AOM/DSS-tumor (T) groups determined via quantitative reverse transcription–polymerase chain reaction (qRT-PCR; *n =* 3). (**F**) Immunohistochemistry (IHC) showed that VDUP1 protein was expressed in the colon of control mice but exhibited lower expression levels in the tumors and adjacent tissues of AOM/DSS-treated mice. 400× magnification. Scale bar = 100 μm. Data are expressed as the mean ± standard error of the mean (SEM). * *p* < 0.05; ** *p* < 0.01; and **** *p* < 0.0001; ns, not significant. UC, ulcerative colitis; HR, hazard ratio; NT, nontumor; T, tumor; OS, overall survival.

**Figure 2 cancers-16-02934-f002:**
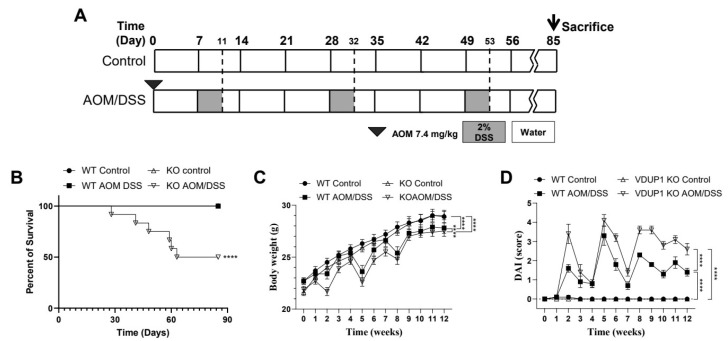
VDUP1 deficiency exacerbates the disease severity in a mouse model of colitis-associated CRC (CAC). (**A**) Schematic representation of the AOM/DSS administration regime to induce tumor progression. WT and *VDUP1* KO mice were intraperitoneally (IP) injected with 7.4 mg/kg AOM on day 0, followed by three cycles of 2% DSS (W/V). (**B**) Survival curve comparing WT versus *VDUP1* KO mice (*n* = 12). Mice were evaluated weekly for (**C**) body weight changes. (**D**) Disease activity index (DAI) was analyzed based on the indicated criteria (Appendix A; *n* = 12) as a sum of body weight change score (Figure 1A), stool consistency score (Figure 1B), and rectal bleeding score (Figure 1C). Data are represented as the mean ± SEM. **** *p* < 0.0001.

**Figure 3 cancers-16-02934-f003:**
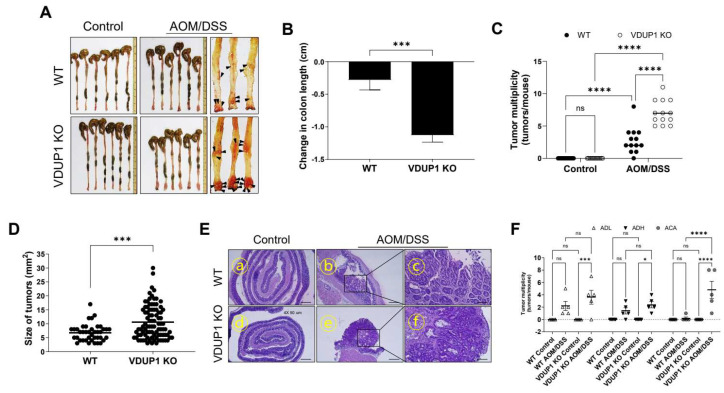
VDUP1 deficiency increases tumor incidence and severity in AOM/DSS-induced mice. WT (*n* = 13) and *VDUP1* KO (*n* = 13) mice were intraperitoneally injected with 7.4 mg/kg AOM on day 0, followed by three cycles of 2% DSS (W/V). On day 85, colons were resected and representative photographs of (**A**) colon are shown. The arrows indicate the tumor. (**B**) Changes in colon length, (**C**) tumor multiplicity (*n* = 13), and (**D**) size of tumors (*n* = 13) were calculated. Representative photographs of (**E**) hematoxylin and eosin (H&E) staining are shown and (**F**) tumor grade (*n* = 5) was analyzed. Scale bars = 500 µm for panels (**E**)—ⓐ, ⓑ, ⓓ, and ⓔ; 100 µm for panels (**E**)—ⓒ and ⓕ. Data are represented as the mean ± SEM. * *p* < 0.05, *** *p* < 0.001, and **** *p* < 0.0001; ns, not significant. ACL, adenoma with low-grade dysplasia; ADH, adenoma with high-grade dysplasia; ACA, adenocarcinoma.

**Figure 4 cancers-16-02934-f004:**
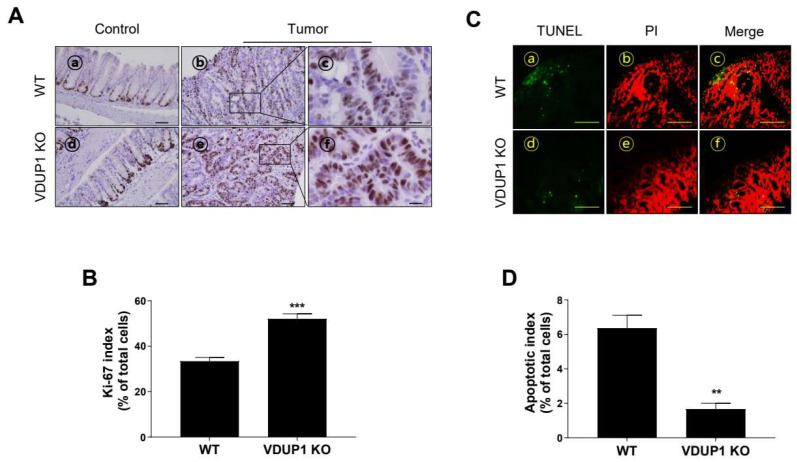
VDUP1 deficiency promotes tumor cell proliferation and inhibits apoptosis in a mouse model of CAC. WT and *VDUP1* KO mice were intraperitoneally injected with 7.4 mg/kg AOM on day 0, followed by three cycles of 2% DSS (*w*/*v*). On day 85, the colons were resected. (**A**) IHC staining of Ki-67 was conducted and (**B**) Ki-67 index was calculated in three different high-power fields for each tissue section (*n* = 3). (**C**) Representative photograph of TdT-mediated dUTP nick-end labeling (TUNEL) staining. Green, TUNEL-positive cells; red, PI-positive cells. (**D**) Apoptotic index was calculated in three different high-power fields for each tissue section (*n* = 3). Scale bars = 50 µm for panel (**A**)—ⓐ, ⓑ, ⓓ, and ⓔ; 200 µm for panel (**A**)—ⓒ, ⓕ, and panel (**C**). Data are represented as the mean ± SEM. For panels (**B**,**D**) ** *p* < 0.01 and *** *p* < 0.001.

**Figure 5 cancers-16-02934-f005:**
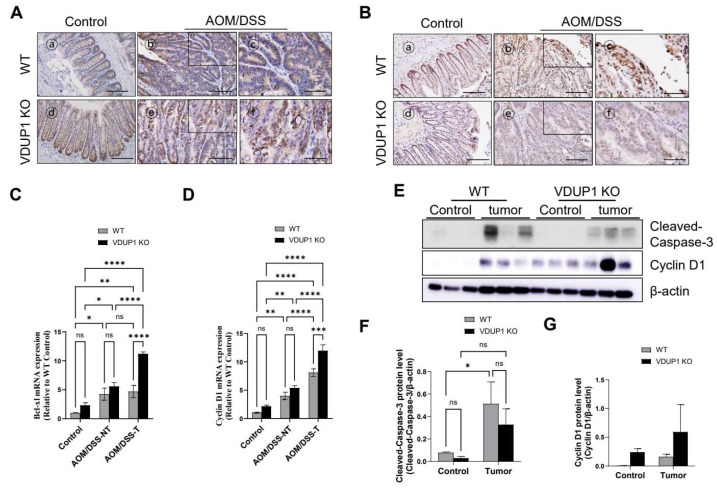
VDUP1 deficiency promotes the signaling pathways associated with cell proliferation and apoptosis inhibition in AOM/DSS-induced CAC. WT and *VDUP1* KO mice were intraperitoneally injected with 7.4 mg/kg AOM on day 0, followed by three cycles of 2% DSS (*w*/*v*). On day 85, the tumors were resected. IHC staining for (**A**) β-catenin and (**B**) p53 was conducted. mRNA levels of (**C**) *Bcl-xl* and (**D**) cyclin D1 were determined via qRT-PCR (*n* = 4). (**E**) Protein levels of cleaved caspase-3 and cyclin D1 were determined via Western blotting (*n* = 3), and staining intensity of (**F**) cleaved caspase-3 and (**G**) cyclin D1 were calculated using the ImageJ software. The uncropped blots are shown in Appendix A. β-actin was used as an internal control. Scale bars = 200 µm for panel (**A**)—ⓐ, ⓑ, ⓓ, ⓔ, for panel (**B**)—ⓐ, ⓑ, ⓓ, and ⓔ; 100 µm for panel (**A**)—ⓒ, ⓕ, for panel (**B**)—ⓒ, and ⓕ. Data are represented as the mean ± SEM. * *p* < 0.05; ** *p* < 0.01, *** *p* < 0.001, and **** *p* < 0.0001; ns, not significant. NT, nontumor; T, tumor.

**Figure 6 cancers-16-02934-f006:**
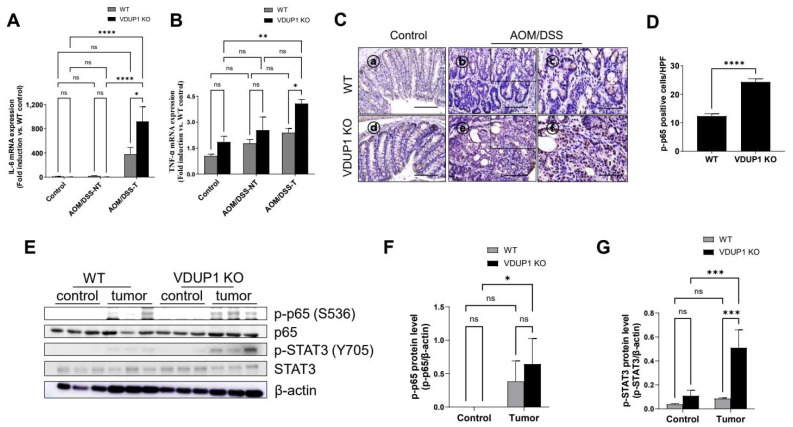
VDUP1 deficiency increases the protein levels of tumor-promoting signaling molecules in a mouse model of CAC. WT and *VDUP1* KO mice were intraperitoneally injected with 7.4 mg/kg AOM on day 0, followed by three cycles of 2% DSS (W/V). On day 85, the tumors were resected. mRNA levels of (**A**) interleukin (IL)-6 and (**B**) tumor necrosis factor-alpha (TNF-α) were determined via qRT-PCR (*n* = 4). (**C**) IHC staining of the active form of p65 was conducted. (**D**) Nuclear incorporation of the active form of p65 was observed in three different high-power fields for each tissue section (*n* = 4). (**E**) Protein levels of p65, STAT3, p-p65, and p-STAT3 were determined via Western blotting, and staining intensities of (**F**) p-p65 and (**G**) p-STAT3 were determined using the ImageJ software. The uncropped blots are shown in Appendix A. β-actin was used as an internal control. Scale bars = 200 µm for panel (**C**)—ⓐ, ⓑ, ⓓ, and ⓔ; 100 µm for panel (**C**)—ⓒ, ⓕ. Data are represented as the mean ± SEM. * *p* < 0.05, ** *p* < 0.01, *** *p* < 0.001, and **** *p* < 0.0001; ns, not significant. NT, nontumor; T, tumor; HPF, high-power field.

## Data Availability

Gene Expression Profiling Interactive Analysis 2 (http://gepia2.cancer-pku.cn/#analysis (accessed on 2 January 2024)) was used to profile VDUP1 expression levels in normal and tumor tissues from The Cancer Genome Atlas-colon adenocarcinoma (TCGA-COAD) dataset. ShinyGEO (http://gdancik.github.io/shinyGEO/ (accessed on 2 January 2024)) was used to analyze VDUP1 levels in UC-non-tumor, UC-dysplasia, and UC-cancer samples from the GSE3629 dataset and disease-specific survival from the GSE38832 dataset. cBioPortal (http://cbioportal.org) was used for overall survival (OS) analysis of the Sidra-LUMC AC-ICAM dataset.

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
