# Peer review of "Vitamin D3 Upregulated Protein 1 Deficiency Promotes Azoxymethane/Dextran Sulfate Sodium-Induced Colorectal Carcinogenesis in Mice"

_cancers, 2024, doi:10.3390/cancers16172934_

Round 1

Reviewer 1 Report

Comments and Suggestions for Authors

The concerns regarding the manuscript are as follows” Vitamin D3 Upregulated Protein Deficiency Promotes Azoxymethane/Dextran Sulfate Sodium-Induced Colorectal Carcinogenesis in Mice”

1.       Even though the work is interesting, the manuscript shows 43% match in iThenticate. Authors should rectify it.

2.       How does VDUP1 affects tumor and normal cell proliferation? The mechanism is not clear from the manuscript. Authors need to highlight it.

3.       In 4A, we are seeing VDUP1 KO in control condition is leading to enhanced proliferation. Again the colon size is reducing in 3A, B. How will the authors clarify this contradiction

4.       4C: Mention what does the green and red color stain indicate

5.       For Fig5, authors should probe for anti-apoptotic proteins like Bcl2 etc

6.       Authors should provide all raw uncropped unprocessed blots clearly indicating from where they are cut and also protein MW.

Author Response

Response to Reviewer 1 Comments

1. Summary

Thank you very much for taking the time to review this manuscript. We appreciate your valuable feedback. Please refer to the following details, as well as the revised manuscript and supplementary information, for a detailed response to your comments.

2. Point-by-point response to Comments and Suggestions for Authors

Comments 1: Even though the work is interesting, the manuscript shows 43% match in iThenticate. Authors should rectify it.

Response 1: We highly appreciate your suggestion regarding the 43% match in iThenticate. We have revised the manuscript to address the issues of overlap and have confirmed the improvements using iThenticate.

We have revised lines 53-61, 86-91, 100-109, 113-120, 126-144, 148-157, 159-180, 182-188, and 191-194 in the manuscript.

Comments 2: How does VDUP1 affects tumor and normal cell proliferation? The mechanism is not clear from the manuscript. Authors need to highlight it.

Response 2: We appreciate your suggestion regarding the question, “How does VDUP1 affect tumor and normal cell proliferation? The mechanism is not clear.”.

We have carefully considered your suggestion and have addressed this by incorporating revisions on lines 284-285 and 381-385.

Comments 3: In 4A, we are seeing VDUP1 KO in control condition is leading to enhanced proliferation. Again the colon size is reducing in 3A, B. How will the authors clarify this contradiction

Response 3: We appreciate your opinion that VDUP1 KO in control condition seems to enhance proliferation. We have carefully considered your suggestion. We measured the colon length and counted the Ki-67 positive cells in the villi of the control groups of WT and VDUP1 KO mice. We did not find any significant differences between WT and VDUP1 KO mice. The data is shown in Fig. S4 and S5 of the supplementary information.

Comments 4: 4C: Mention what does the green and red color stain indicate

Response 4: Thank you for your detailed comments on the interpretation of the Fig. 4C.

We have carefully considered your suggestion. We have specified the content “Green, TUNEL-positive cells; Red, PI-positive cells.” On page 7, line 295 of the manuscript.

Comments 5: For Fig5, authors should probe for anti-apoptotic proteins like Bcl2 etc.

Response 5: We appreciate your suggestion regarding probing for anti-apoptotic proteins like Bcl2 in Fig 5. We aimed to elucidate the role of VDUP1 in CAC from a broader perspective. In this study, we concluded that VDUP1 plays a crucial role in inducing apoptosis within CAC tissues through TUNEL staining, detection of cleaved-Caspase-3 protein, Bcl-xl PCR, and examination of upstream signaling molecules such as p53, β-catenin, p-p65, and p-STAT3. We are continuously researching the role of the VDUP1 gene and its potential as a drug target, and we will certainly incorporate your suggestion in our upcoming in-depth studies in CAC.

Comments 6: Authors should provide all raw uncropped unprocessed blots clearly indicating from where they are cut and also protein MW.

Response 6: We appreciate your opinion regarding “Authors should provide all raw uncropped unprocessed blots clearly indicating from where they are cut and also protein MW.”

We have carefully considered your suggestions. We have included unprocessed western blot images with loading lines of size marker in supplementary information as Fig. S2 and S3.

Reviewer 2 Report

Comments and Suggestions for Authors

This study determined the expression of VDUP1 in patients with CRC and CAC. According to these researchers, in searches on multiple independent datasets, there has been a consistent downregulation of VDUP1 with associated poor prognosis. Experimental validation in animal models further supports these findings, relating VDUP1 deficiency to increased tumorigenesis and inflammation. The results highlight that VDUP1 could be a potential prognostic marker and therapeutic target for such cancers; further studies are needed to validate its mechanistic and clinical aspects. Some comments are listed below for the authors' attention.

Introduction

1. The introduction would benefit from better integration with previous literature.

2. The limitations of this study are acknowledged and discussed.

3. The introduction could be much clearer for differentiating between general CRC and colitis-associated CRC (CAC). This distinction is clearly drawn on page itself.

4. The transition from general CRC data to focus on chronic inflammation and genetic alterations may be smoother.

Methods:

5. The induction procedure of colitis and CAC in mice describes the dosing of AOM and DSS. The authors were asked to include a statement on the records of disease activity index parameters and animal survival.

6. Homogenization details: Comment on homogenizer type and process duration.

7. All information regarding primer design and validation guarantees the specificity and efficiency of the qRT-PCR approach.

8. The methods for histological examination, IHC, and TUNEL assays are explained.

9. The details of the membrane blocking and washing steps should be provided to guarantee reproducibility.

10. Describe details of the validation of antibody specificity and the method used for quantifying the band intensities.

Results and discussion

11. Such findings are provided in the form of figures and quantitative data, thus making it easy to directly comprehend the relationship between VDUP1 expression and cancer development.

12. The research only involved a few animal and human tissue samples, which may impinge on generalizability. I meant from a small sample size, such conclusions cannot be drawn.

13. Although the effect on survival analysis was significant, a longer follow-up period would yield more accurate results and further confirm the effect of VDUP1 deficiency on long-term survival and disease progression.

14. Some datasets showed high variability in the expression level of VDUP1. Once again, harmonization of the measurement techniques or studies on more homogeneous populations of samples will help reduce this variability. 15. The study did not fully account for the potential confounding factors of genetic background in mice or environmental factors that may have impacted the results. Controls for these variables would provide better insight into the data. 

Author Response

Response to Reviewer 2 Comments

1. Summary

Thank you very much for taking the time to review this manuscript. We appreciate your valuable feedback. Please refer to the following details, as well as the revised manuscript and supplementary information, for a detailed response to your comments.

2. Point-by-point response to Comments and Suggestions for Authors

Comments 1: The introduction would benefit from better integration with previous literature.

Response 1: We appreciate your opinion that the introduction would benefit from better integration with previous literature, and we have reconsidered this aspect accordingly.

Comments 2: The limitations of this study are acknowledged and discussed.

Response 2: We appreciate your opinion that the limitations of this study are acknowledged and discussed. It has been helpful for further consideration of our research.

Comments 3: The introduction could be much clearer for differentiating between general CRC and colitis-associated CRC (CAC). This distinction is clearly drawn on the page itself.

Response 3: We appreciate your opinion on the distinction between CRC and CAC. We have made every effort to incorporate your feedback throughout the introduction.

Comments 4: The transition from general CRC data to focus on chronic inflammation and genetic alterations may be smoother.

Response 4: We appreciate your opinion on “The transition from general CRC data to focus on chronic inflammation and genetic alterations may be smoother.”. We have carefully considered your suggestion and have strived to incorporate the molecular biological phenomena identified in CRC and CAC to date

Comments 5: The induction procedure of colitis and CAC in mice describes the dosing of AOM and DSS. The authors were asked to include a statement on the records of disease activity index parameters and animal survival.

Response 5: We appreciate your opinion on including a statement on including a statement on the records of disease activity index parameters and animal survival.

We have carefully considered your suggestions and the information is provided in Table S1 and  Fig. S6 of the supplementary information.

Comments 6: Homogenization details: Comment on homogenizer type and process duration.

Response 6: We appreciate your opinion on homogenization details: comment on homogenizer type and process duration.

We have carefully considered your suggestions, and have included this information on page 3, lines 124-125 of the manuscript.

Comments 7: All information regarding primer design and validation guarantees the specificity and efficiency of the qRT-PCR approach.

Response 7: We appreciate your opinion on presenting all information regarding primer design and validation to guarantee the specificity and efficiency of the qRT-PCR approach. We have carefully considered your suggestion, and the information is provided in Table S2 of the supplementary information.

Comments 8: The methods for histological examination, IHC, and TUNEL assays are explained.

Response 8: We appreciate your positive feedback on the methods for histological examination, IHC, and TUNEL assays.

Comments 9: The details of the membrane blocking and washing steps should be provided to guarantee reproducibility.

Response 9: We appreciate your opinion on the details of the western immunoblotting analysis.

We have carefully considered your suggestions, and have included this information on page 4, lines 159-180 of the manuscript

Comments 10: Describe details of the validation of antibody specificity and the method used for quantifying the band intensities.

Response 10: We appreciate your opinion on the need to provide the methods for antibody specificity and band intensity measurements. We have carefully considered your suggestion and included the method for measuring band intensities on page 4, 180 line of the manuscript, and information regarding antibody specificity is provided in Table S3 of the supplementary information.

Comments 11: Such findings are provided in the form of figures and quantitative data, thus making it easy to directly comprehend the relationship between VDUP1 expression and cancer development.

Response 11: We appreciate your positive feedback regarding the data processing and presentation.

Comments 12: The research only involved a few animal and human tissue samples, which may impinge on generalizability. I meant from a small sample size, such conclusions cannot be drawn.

Response 12: We appreciate your opinion that the research only involved a few animal and human tissue samples, which may impinge on generalizability.

We conducted experiments and analyses at a level that allows for statistical significance through public data analysis and animal experiments. Specifically, for the animal experiments, we made efforts to adhere to the 3R principles.

Comments 13: Although the effect on survival analysis was significant, a longer follow-up period would yield more accurate results and further confirm the effect of VDUP1 deficiency on long-term survival and disease progression.

Response 13: We appreciate your opinion regarding the survival analysis. We have carefully considered your suggestion. In fact, we monitored the survival of WT and VDUP1 KO mice not only with AOM/DSS but also with DSS alone until day 140. In WT animals, there were no deaths in either the DSS alone or the AOM/DSS groups. VDUP1 KO animals showed a statistically significant mortality rate in the AOM/DSS group, while the DSS alone group had deaths, but without statistical significance. Since there were no changes in survival rates from day 80 to day 140, we included the results up to day 80 in the manuscript, and the data up to day 140 are presented in Fig. S6 of the supplementary information.

Comments 14: Some datasets showed high variability in the expression level of VDUP1. Once again, harmonization of the measurement techniques or studies on more homogeneous populations of samples will help reduce this variability.

Response 14: We appreciate your opinion on considering the environmental factors for experimental animals. We have carefully considered your feedback. In this study, we have utilized public data from TCGA, GEO, and cBioPortal to derive objective and reliable analysis results. We will incorporate your suggestions and strive to analyze all possible techniques and cohorts in future research.

Comments 15: The study did not fully account for the potential confounding factors of genetic background in mice or environmental factors that may have impacted the results. Controls for these variables would provide better insight into the data. 

Response 15: We appreciate your opinion regarding potential confounding factors of genetic background in mice or environmental factors. We have maintained the highest standards in the management of laboratory animals regarding environmental factors, but the potential results derived from the genetic background of the VDUP1 KO mice have not yet been fully elucidated. Therefore, we will actively incorporate your feedback and strive to provide a thorough understanding of the possible variations due to genetic background in future research.

Round 2

Reviewer 2 Report

Comments and Suggestions for Authors

No further comment. Thank you !